# Antibiofilm Activities of Multiple Halogenated Pyrimidines Against *Staphylococcus aureus*

**DOI:** 10.3390/ijms252312830

**Published:** 2024-11-28

**Authors:** MinHwi Sim, Yong-Guy Kim, Jin-Hyung Lee, Jintae Lee

**Affiliations:** School of Chemical Engineering, Yeungnam University, Gyeongsan 38541, Republic of Korea; tla7686@yu.ac.kr (M.S.); yongguy7@ynu.ac.kr (Y.-G.K.)

**Keywords:** antibiofilm, antimicrobial, halogenated pyrimidines, hemolysis, *Staphylococcus aureus*

## Abstract

*Staphylococcus aureus*, prevalent in hospital and community settings, forms biofilms that are highly resistant to antibiotics and immune responses, complicating treatment and contributing to chronic infections. These challenges underscore the need for novel treatments that target biofilm formation and effectively reduce bacterial virulence. This study investigates the antibiofilm and antimicrobial efficacy of novel halogenated pyrimidine derivatives against *S. aureus*, focusing on three compounds identified as potent biofilm inhibitors: 2,4-dichloro-5-fluoropyrimidine (24DC5FP), 5-bromo-2,4-dichloro-7H-pyrrolo[2,3-d]pyrimidine (24DC5BPP), and 2,4-dichloro-5-iodo-7H-pyrrolo[2,3-d]pyrimidine (24DC5IPP). The three active compounds are bacteriostatic. In particular, 24DC5FP at 5 µg/mL achieved a 95% reduction in hemolysis with a minimum inhibitory concentration (MIC) of 50 µg/mL. Interestingly, 24DC5FP increased cell size and produced wrinkled colonies. qRT-PCR analysis showed that 24DC5FP suppressed the gene expressions of *agrA* and *RNAIII* (quorum sensing regulator and effector), *hla* (α-hemolysin), *nuc1* (nucleases nuc1), and *saeR* (*S. aureus* virulence regulator). These findings suggest that extensive halogenation enhances the antibiofilm and antivirulence activities of pyrimidine derivatives, offering a promising strategy for combatting *S. aureus* infections, including those resistant to conventional treatments.

## 1. Introduction

*Staphylococcus aureus*, a critical pathogen in both nosocomial and community settings, presents significant therapeutic challenges due to its propensity to form biofilms [1,2]. These biofilms are complex communities of bacteria that are highly resistant to both antibiotics and host immune responses, making infections difficult to eradicate and often chronic [3]. This resilience is compounded in strains such as methicillin-resistant *S. aureus* (MRSA), where traditional antimicrobial strategies frequently fail, underscoring the urgent need for innovative treatment approaches. Recent advances in microbial management have highlighted the potential of targeting biofilm-associated infections with agents that inhibit biofilm formation and reduce virulence without promoting resistance [4].

The current study explores the efficacy of novel antibiofilm and antimicrobial compounds, specifically halogenated pyrimidine derivatives. Recent studies suggest that halogenated compounds could enhance target binding affinity for improved antimicrobial effects while simultaneously reducing toxicity [5,6]. Moreover, heterocyclic pyrimidine scaffolds have shown a broad range of biological activities, encompassing antioxidant, antimicrobial, antiviral, anti-inflammatory, and anticancer properties [7,8].

Earlier studies suggest that several halogenated compounds exhibit antimicrobial and antibiofilm properties against *S. aureus*. For example, halogenated indoles [9], halogenated phenazines [10,11,12], halogenated quinolines [13], and iodinated hydrocarbons [14] have been shown to inhibit *S. aureus* biofilm formation. Recently, multisubstituted pyrimidines [15] and halogenated pyrrolopyrimidines [16] were found to inhibit bacterial cell growth and biofilm formation of *S. aureus*.

Therefore, this research investigated the antimicrobial and antibiofilm capabilities of various halogenated pyrimidine or pyrrolopyrimidine analogs against *S. aureus*, identifying three multi-halogenated pyrimidines in the process. Particularly, their heterocyclic pyrimidine and pyrrolopyrimidine scaffolds are known for exhibiting diverse biological activities [17,18], making them promising candidates for biofilm inhibition. This study employed methods such as killing dynamics, microscopy, slime production and hemolysis assays, and transcriptomic analysis to uncover the mechanisms behind the compounds’ efficacy.

## 2. Results

### 2.1. Antimicrobial and Antibiofilm Effects of Diverse Halogenated Pyrimidines Against S. aureus

The preliminary screening of antibiofilm effects of pyrimidine and 30 halogenated pyrimidine derivatives on *S. aureus* was conducted using 96-well plates at concentrations of 50 µg/mL. The halogenated derivatives showed varied levels of biofilm inhibition on *S. aureus* (Figure 1). Comprehensive results, including the chemical structures of these compounds, are detailed in Figure 1. Notably, certain compounds, such as 2,4-dichloro-5-fluoropyrimidine (#9, referred to as 24DC5FP), 5-bromo-2,4-dichloro-7H-pyrrolo[2,3-d]pyrimidine (#25, referred to as 24DC5BPP), and 2,4-dichloro-5-iodo-7H-pyrrolo[2,3-d]pyrimidine (#31, referred to as 24DC5IPP), demonstrated significant biofilm inhibition at 50 µg/mL. In contrast, 7H-pyrrolo[2,3-d]pyrimidine (#13) and pyrimidine (#32) did not exhibit any inhibitory effect (Figure 1). Specifically, compounds #25 (24DC5BPP) and #31 (24DC5IPP) reduced *S. aureus* biofilm formation by over 90% at 50 µg/mL. It is notable that the more effective compounds each contained three halogen atoms, such as fluorine, bromine, dichlorine, or iodine, whereas less active halogenated derivatives typically had one or two halogen atoms.

The minimum inhibitory concentrations (MICs) of various pyrimidine derivatives were evaluated to measure their antibacterial efficacy. MICs for 24DC5FP, 24DC5BPP, and 24DC5IPP were recorded at 50, 50, and 100 µg/mL, respectively (Figure 1). Conversely, 4-chloro-5-iodopyrimidine (#2) exhibited an MIC of 200 µg/mL, while other compounds had MICs exceeding 400 µg/mL. These results indicate that the biofilm suppression by 24DC5FP, 24DC5BPP, and 24DC5IPP may be linked to their antibacterial capabilities. This research provides the initial documentation of the antimicrobial and antibiofilm effectiveness of 24DC5FP, 24DC5BPP, and 24DC5IPP against *S. aureus*. These three compounds were subsequently selected for deeper exploration of their potential antibiofilm and antivirulence properties.

The antimicrobial effects of the three compounds were further investigated. Each exhibited a concentration-dependent reduction in planktonic cell growth, with MICs ranging from 50 to 100 µg/mL, and the cell growth measurement was conducted up to 50 µg/mL (Figure 2A–C). A time-kill kinetic analysis was performed to assess their bacteriostatic or bactericidal impacts on *S. aureus.* The findings showed that all three compounds functioned in a bacteriostatic capacity, with 24DC5BPP and 24DC5IPP maintaining a cell count of 10^7^ with a 200 µg/mL dosage after 24 h, while 24DC5FP reduced the cell count to 10^5^ (Figure 2D–F).

### 2.2. Antibiofilm Effects of Dichloro-Pyrimidines

A detailed biofilm assay revealed that the three compounds exhibited dose-dependent biofilm inhibition against *S. aureus* (Figure 3A–C). Their antibiofilm capabilities were evaluated through live microscopy and SEM. Live microscopic imaging demonstrated that at concentrations of 50 or 100 µg/mL, these active compounds significantly prevented biofilm formation, in contrast to the dense biofilms observed in untreated samples (Figure 3D). SEM studies further validated the antibiofilm efficacy of these compounds, showing fewer cells in treated samples compared with controls (Figure 3E). Notably, treatment with 24DC5FP resulted in increased cell size, whereas the other two compounds did not significantly alter the morphology of *S. aureus* cells.

### 2.3. Impact of Dichloro-pyrimidines on Slime Formation and Hemolysis in S. aureus

To understand how dichloro-pyrimidines inhibited *S. aureus* biofilm formation, investigations into slime production and hemolytic activity were conducted. On Congo red agar plates, 24DC5FP led to lower slime production while 24DC5BPP and 24DC5IPP did not significantly change slime production (Figure 4A). Additionally, colony morphology was observed in the presence of dichloro-pyrimidine derivatives. 24DC5FP produced wrinkled colonies, whereas 24DC5BPP and 24DC5IPP showed similar colony morphology to untreated samples (Figure 4B). The pellet color of the 24DC5BPP treatment group showed a slight difference compared with the none, suggesting a minor effect on staphyloxanthin production. However, the three dichloro-pyrimidine derivatives did not induce substantial changes in staphyloxanthin production overall (Figure 4C).

Hemolytic activity, primarily influenced by alpha-hemolysin, represents a critical virulence attribute of *S. aureus* [19,20]. This toxin, produced from the *hla* gene, can disrupt red blood cells and positively contribute to biofilm formation [21]. Hence, we assessed the impact of halogenated pyrimidines on the hemolytic capacity of *S. aureus*. Notably, the smaller molecule, 24DC5FP, was found to suppress hemolysis in a concentration-dependent manner, achieving more than a 95% reduction at as low as 5 μg/mL (Figure 4D). Conversely, neither 24DC5BPP nor 24DC5IPP demonstrated any hemolytic suppression at concentrations up to 20 μg/mL (Figure 4E,F). In contrast to the MIC results, the hemolysis assay conducted in a shaking incubator revealed that the suppression of hemolysis was more prominent.

For comparison purposes, the antibiofilm and anti-hemolysis activities of gentamicin were tested. The MIC of gentamicin against *S. aureus* was 20 µg/mL (Appendix A). As expected, gentamicin dose-dependently inhibited *S. aureus* biofilm formation, while gentamicin up to 100 µg/mL could not completely inhibit it (Appendix A). Also, gentamicin dose-dependently inhibited the hemolytic activity, mainly due to the growth inhibition (Appendix A). It appears that the antibiofilm and anti-hemolysis activities of 24DC5FP are similar to those of gentamicin.

### 2.4. Impact of 24DC5FP on Gene Expression in S. aureus 

To understand the molecular mechanisms of 24DC5FP in *S. aureus*, qRT-PCR analysis was used to examine changes in the expression of 11 biofilm-related genes in *S. aureus* cells. The primer sequences used in this study were adopted from a previously published study [22]. Treatment with 24DC5FP (50 µg/mL) led to a significant downregulation of the gene expression of *agrA* (quorum sensing regulator), *hla* (α-hemolysin), *nuc1* (nuclease 1), *RNAIII* (quorum sensing regulator), and *saeR* (virulence regulator), whereas the other genes tested were relatively less affected (Figure 4G). Interestingly, the expression levels of *agrA* and *hla* were reduced by 12- and 146-fold, respectively, upon treatment with 24DC5FP, demonstrating its potential antibiofilm and antivirulence properties.

## 3. Discussion

Antibiotic resistance and chronic infections are major challenges associated with biofilm formation, highlighting the growing importance of biofilm research [23]. The present study reports the antimicrobial and antibiofilm effects of multi-halogenated pyrimidine and pyrrolopyrimidines against *S. aureus*, shedding light on some of their mechanisms of action. Notably, three active dichloro-pyrimidine derivatives, 24DC5FP, 24DC5BPP, and 24DC5IPP (Figure 1), each contain three halogen atoms—fluorine, bromine, dichlorine, or iodine. This suggests that extensive halogenation enhances these activities.

Heterocyclic pyrimidine or pyrrolopyrimidine structures exhibit a wide array of biological functions [7,8], including antimicrobial and antibiofilm activities of halogenated indoles [9], halogenated phenazines [10,11,12], halogenated quinolines [13], iodinated hydrocarbons [14], multisubstituted pyrimidines [15], and halogenated pyrrolopyrimidines [16]. This study is the first report highlighting the significance of extensive halogenation in enhancing their activity.

The distinction in the mechanisms of action observed between these compounds reflects their unique structural features. Specifically, 24DC5FP primarily inhibits planktonic cell and alpha-hemolysin production (Figure 2 and Figure 4D), and 24DC5BPP and 24DC5IPP partially reduce cell growth (Figure 2). This differentiation likely arises from the specific halogenation patterns in their structures. Previously, halogenation substitution with brome and chloride at the C4 position of pyrimidines enhanced the antibiofilm activity [15], and amine substitution with halogenated benzene at the C4 position of pyrrolopyrimidines increased the antimicrobial activity [15] against *S. aureus*. Additionally, the three compounds in this study feature two chloride atoms at the C2 and C4 positions, along with another halogen atom of fluorine, bromine, or iodine (Figure 1). These findings suggest that specific halogenation patterns influence bacterial interactions and may also contribute to their distinct antibiofilm and antimicrobial activities. 

Interestingly, the treatment of 24DC5FP caused wrinkled colonies (Figure 4A,B) and increased cell size (Figure 3E), which is an unusual morphology observation in *S. aureus* study. It is well known that *S. aureus* small-colony variants are associated with the development of persistent infections while the production of virulence factors is often reduced [24,25]. While wrinkled or rugose bacterial colonies on solid agar media are a common biofilm phenotype [26], wrinkled *S. aureus* colonies have been rarely reported. A mutant study showed that deletion of *agr*, *purA*, and *purK* developed small and wrinkled colonies [27] and suggested that purine synthesis may affect macrocolony morphology. Hence, 24DC5FP, having pyrimidine and three halogen atoms, may affect or target the biosynthesis of purine, having pyrimidine and imidazole, which leads to reduced cell growth. 

Dramatic increase of cell size in *S. aureus* is also an unusual phenotypic change by the treatment of 24DC5FP (Figure 3E). FtsZ is an essential protein for cell division in most bacteria, including *S. aureus* [28], and several antibiotics targeting peptidoglycan synthesis arrest cell division [29,30]. However, the impact of 24DC5FP on cell shape appears to be different from the inhibition of cell division since SEM images did not show septum constriction. Since the deletion of *ugtP* (encoding an enzyme for glycolipid anchor) in *S. aureus* led to large swollen cells [31], which is similar to our observation (Figure 3E). Hence, 24DC5FP may affect the production of lipoteichoic acid, which is another important component of the *S. aureus* cell envelope.

Notably, 24DC5FP repressed the gene expression of *agrA*, *hla*, *nuc1*, *RNAIII*, and *saeR,* partially elucidating the mechanism involved (Figure 4G). The quorum-sensing system, encoded by *agr* and *RNAIII*, controls the biofilm formation and virulence factors such as hemolytic activity, proteases, nucleases, and other toxins in *S. aureus* [32,33]. In particular, the expression of α-hemolysin (*hla*) was the most downregulated by 24DC5FP, which directly supports the inhibitory effect of 24DC5FP on hemolytic activity (Figure 4D). Additionally, SaeR (*S. aureus* virulence regulator) directly affects *hla* transcription [34]. These results demonstrate that 24DC5FP suppresses virulence characteristics by repressing *agrA*, *RNAIII*, and *saeR*.

Anti-virulence strategies by small molecules are gaining interest against drug-resistant microbes. Particularly, *S. aureus* can produce virulence factors, such as various adhesins, alpha-toxin (Hla), staphyloxanthin, and enterotoxins [35]. Notably, 24DC5FP at 5 µg/mL (one tenth of MIC) inhibited hemolytic activity (Figure 4D). α-Hemolysin is a major toxin in *S. aureus* [35], and the hemolysin is required for biofilm formation by *S. aureus* [21]. Since 2020, it has been reported that various natural and synthetic small molecules inhibited biofilm formation as well as α-hemolysin production [22,36,37,38,39,40,41,42,43,44,45,46,47,48,49,50,51,52]. Therefore, these results strongly support the positive relationship between *hla* and biofilm formation in *S. aureus*.

Halogenation of small molecules could improve antimicrobial and antibiofilm activities and is a promising strategy against diverse drug-resistant microbes [5,6]. Recently, halogenated pyrrolopyrimidines showed antimicrobial and antibiofilm activities against *S. aureus* and synergistic effects with an antimicrobial peptide [16]. Hence, it is possible that these multiple halogenated pyrimidines would exhibit antimicrobial and antibiofilm activities against other drug-resistant bacteria. Also, these bacteriostatic dichloro-pyrimidines could be used as an adjuvant along with bactericidal antibiotics to diminish drug-resistant biofilm formation and virulence characteristics.

## 4. Materials and Methods

### 4.1. Bacterial Growth Conditions and Chemicals

This investigation employed the *S. aureus* ATCC 6538 strain, obtained from the American Type Culture Collection (Manassas, VA, USA), which was cultured in Luria-Bertani (LB) broth at 37 °C. Pyrimidine and thirty-one halogenated pyrimidine derivatives (purity ≥ 98%) were obtained from Combi Blocks (San Diego, CA, USA), as detailed in Figure 1. Gentamicin was obtained from Sigma-Aldrich, St. Louis, MO, USA. These compounds were solubilized in dimethyl sulfoxide (DMSO). For the control experiments, 0.1% *v*/*v* DMSO was used, showing no significant effect on the growth or biofilm formation of *S. aureus* at this concentration.

For the planktonic cell growth assessment, turbidity and colony-forming units (CFU) of *S. aureus* were evaluated in 96-well plates incubated with or without pyrimidine derivatives for 24 h. In the minimum inhibitory concentration (MIC) analysis, an overnight culture of *S. aureus* was diluted to an OD_600_ of 0.1 (approximately 10^7^ CFU/mL) and exposed to each compound in LB medium for 24 h to assess growth inhibition. The MIC represents the lowest concentration at which no growth of planktonic cells is detectable. These experiments were independently replicated at least twice, with each conducted in triplicate.

### 4.2. Quantitative Biofilm Assessment Using Microtiter Plates

An overnight culture of *S. aureus* was diluted to approximately 10^7^ cells and combined with pyrimidine derivatives in LB medium. Three hundred microliters of this mixture were transferred into 96-well polystyrene plates (SPL Life Sciences, Ansan, Republic of Korea) and incubated at 37 °C for 24 h without shaking. After incubation, the growth of planktonic cells was determined by measuring the optical density at 600 nm (OD_600_) using a Multiskan EX microplate reader (Thermo Fisher Scientific, Waltham, MA, USA). For biofilm quantification, the supernatant was removed, and the wells were thoroughly washed with distilled water. Biofilm cells were stained with 0.1% crystal violet for 20 min, washed, and the dye was dissolved in 95% ethanol. The absorbance of the resulting solution was measured at 570 nm (OD_570_) with the Multiskan EX microplate reader. The average results were calculated from at least six replicates in two separate experiments [53].

### 4.3. Temporal Bactericidal Analysis

The bactericidal or bacteriostatic properties of pyrimidine derivatives were evaluated with slight adjustments [54]. An overnight culture of *S. aureus* (~10^7^ CFU/mL) was added to 2 mL tubes containing the pyrimidine derivatives at concentrations of 100 or 200 µg/mL. These tubes were incubated at 37 °C with agitation at 250 rpm. Samples of 100 µL were collected at intervals of 0, 2, 4, 8, and 24 h, diluted serially, and plated on LB agar. After incubating these plates at 37 °C, the colony-forming units (CFU) were counted, and the results were expressed as log_10_ CFU/mL.

### 4.4. Visual Analysis of Biofilms via Live Microscopy and Scanning Electron Microscopy (SEM)

To evaluate the antibiofilm effects of three halogenated pyrimidine derivatives on *S. aureus*, biofilms were cultivated in 96-well plates for 24 h at 37 °C using concentrations of 0, 50, or 100 µg/mL of the derivatives. After the incubation period, planktonic cells were eliminated by triple washing with distilled water. These biofilms were visualized using the iRiS^TM^ Digital Cell Imaging System (Logos BioSystems, Anyang, Republic of Korea), and the images obtained were transformed into 2D and 3D color-coded representations via ImageJ software v. 1.54 [55].

The SEM analysis was performed following a standard protocol. Initially, 300 µL of *S. aureus* cell suspension (~10^7^ CFU/mL) mixed with pyrimidine derivatives (0, 50, or 200 µg/mL) was added to 96-well plates equipped with sterile nylon filter membranes (0.4 × 0.4 mm). The plates were kept at 37 °C under static conditions for 24 h. After incubation, the biofilms were fixed with 2.5% formaldehyde and 2.5% glutaraldehyde for 24 h and dehydrated using a sequential ethanol gradient. After undergoing critical-point drying with an HCP-2 unit (Hitachi, Tokyo, Japan) and platinum sputter-coating, the samples were visualized under an S-4800 scanning electron microscope (Hitachi) at 15 kV and 10 kV [55].

### 4.5. Slime Assay

Assays for colony morphologies and slime production were performed on Congo red agar (CRA) as outlined previously [55]. The composition of CRA included brain–heart infusion broth (37 g/L), sucrose (36 g/L), agar (15 g/L), and Congo red (0.8 g/L). An amount of 5 µL of overnight S. aureus cultures was applied to CRA plates containing pyrimidine derivatives at concentrations of 50 µg/mL and incubated at 37 °C for 48 h prior to imaging. These assays were conducted in duplicate. The appearance of black-colored colonies was indicative of significant slime production, whereas pale-colored colonies suggested minimal or no slime production.

### 4.6. Colony Morphology on BHI Agar Plate

Assays for colony morphologies were performed on brain–heart infusion agar (BHI agar) as outlined previously. The composition of BHI agar included brain–heart infusion broth (37 g/L), sucrose (36 g/L), and agar (15 g/L). An amount of 5 µL of overnight *S. aureus* cultures was applied to BHI agar plates containing pyrimidine derivatives at concentrations of 50 µg/mL and incubated at 37 °C for 48 h prior to imaging.

### 4.7. Staphyloxanthin Assay

To evaluate the effect of chemicals on staphyloxanthin production in *S. aureus* [14]. Single colonies were inoculated into 14 mL tubes containing 2 mL of LB liquid medium. The tubes were then placed in an incubator (37 °C, 250 rpm shaking) for 15 h. The overnight culture was then diluted at 1:100, and 20 mL of the diluted culture was transferred into a flask for re-inoculation. The culture was then incubated in a shaking incubator at 37 °C for 3 h. After this incubation, each chemical was added to a separate flask at a concentration of 50 µg/mL. Samples were taken at 16 h after chemical addition, and each culture was harvested by centrifugation at 10,000 rpm for 10 min to observe the color of staphyloxanthin. To compensate for growth inhibition, harvested volumes were varied as 3 mL for none, 8 mL for 24DC5FP, 16 mL for 24DC5BPP, and 8 mL for 24DC5IPP, respectively.

### 4.8. Hemolysis Assay

To examine the anti-hemolytic effects of pyrimidine derivatives, the protocol described in a previous study was adapted [55]. A 2 mL *S. aureus* cell suspension (~10^7^ CFU/mL) was placed in 14 mL tubes and exposed to varying concentrations of pyrimidine derivatives (0, 2, 5, 10, and 20 µg/mL) and incubated for 24 h at 37 °C with shaking at 250 rpm. Sheep blood was processed by centrifuging at 3000 rpm for 5 min, washing three times in PBS buffer, and adjusting the final concentration to 3.3% (*v*/*v*) with PBS. After incubation, 100 µL of bacterial culture was mixed with 1 mL of the prepared blood solution and further incubated with shaking at 37 °C for 1 h. The samples were centrifuged at 10,000 rpm for 5 min, and the absorbance of the supernatants was measured at 543 nm (OD543) to quantify hemolysis.

### 4.9. RNA Isolation and qRT-PCR

To investigate the molecular mechanisms affected by the most active compound, 24DC5FP, qRT-PCR was conducted following a previously described protocol with slight modifications [22]. *S. aureus* cells (~10^7^ CFU/mL) were cultured in LB medium (25 mL) at 37 °C with shaking at 250 rpm for 3 h. Afterward, 24DC5FP was added to a final concentration of 50 µg/mL, and the cultures were further incubated for an additional 3 h under identical conditions. RNA integrity was preserved using RNase inhibitor (RNAlater, Ambion, TX, USA), and cell lysis was achieved by bead beating with glass beads (150–212 μm, Sigma-Aldrich). Total RNA was extracted with the Qiagen RNeasy MiniKit (Valencia, CA, USA) following the manufacturer’s instructions. qRT-PCR analysis was performed with SYBR™ Green qPCR Master Mix (Applied Biosystems, Foster City, CA, USA) using the ABI StepOne Real-Time PCR System (Applied Biosystems). The cycle threshold (Ct) values were analyzed, and relative gene expression was determined using the 2^−ΔΔCT^ method with *16S rRNA* as the internal control. Data were collected from two independent cultures, with four technical replicates for each gene. Primers were used as previously described [22].

### 4.10. Statistical Analysis

All experimental procedures were carried out with two independent cultures, each having two to three replicates, and findings are reported as means ± standard deviations (SDs). Statistical significance was determined using the student’s t-test, with a *p*-value of less than 0.05 indicating significant differences.

## 5. Conclusions

The study demonstrates for the first time that three dichloro-pyrimidines exhibited high antibiofilm activity along with medium antimicrobial activity against *S. aureus*. In particular, 24DC5FP displayed strong anti-hemolysis activity. It appears that the modes of action between the three hits are different and should be further identified. Further molecular studies to identify specific targets, such as genes or proteins, are required to elucidate the precise mechanisms of action. Additionally, in vivo and toxicological studies are necessary to confirm the efficacy and safety of these compounds for potential clinical applications.

## Figures and Tables

**Figure 1 ijms-25-12830-f001:**
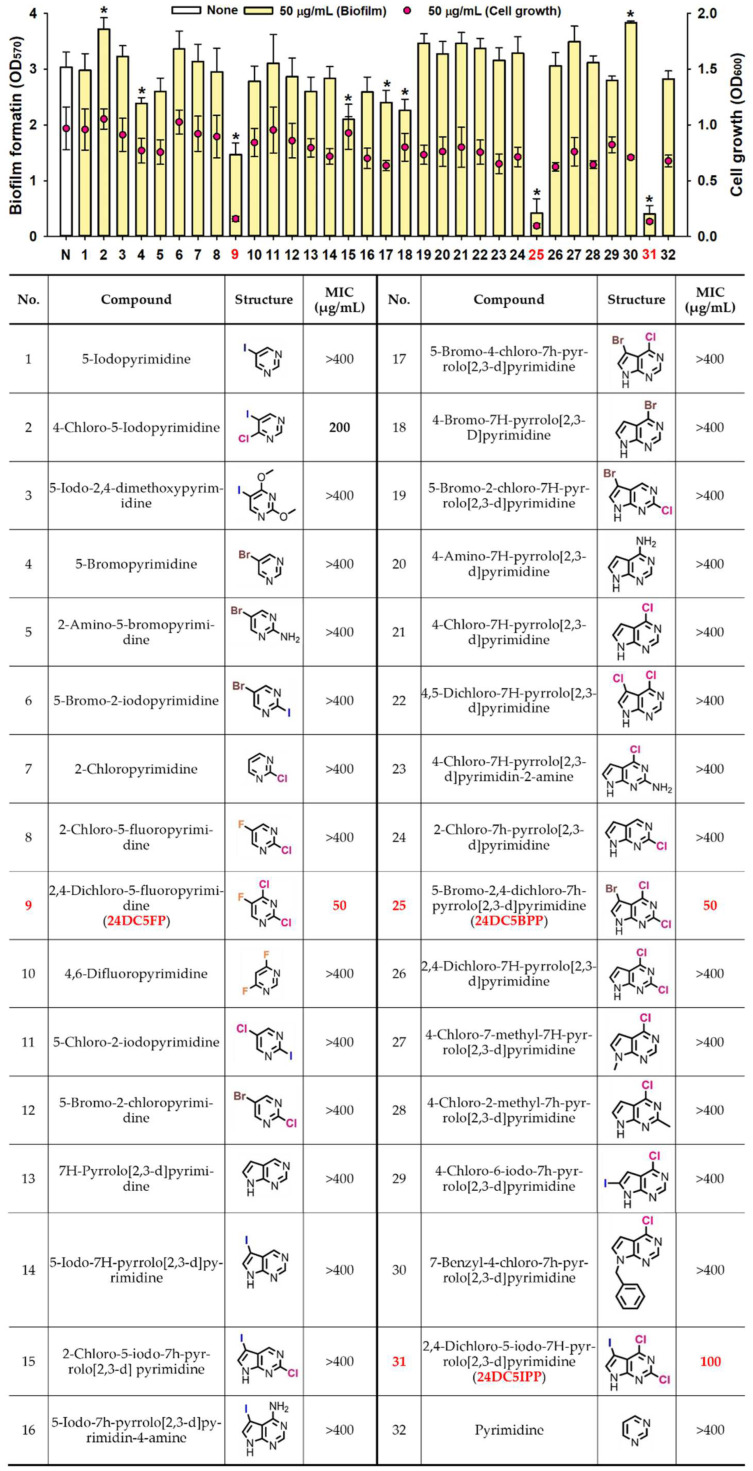
The antibiofilm and antibacterial screening of various pyrimidine derivatives. Biofilm formation by *S. aureus* with pyrimidine derivatives at 50 µg/mL in 96-well polystyrene plates after 24 h culture. Asterisks (*) indicate significant differences in biofilm formation (*p* < 0.05), and error bars display the standard deviation. The listed numbers correspond to the chemical names and their respective structures.

**Figure 2 ijms-25-12830-f002:**
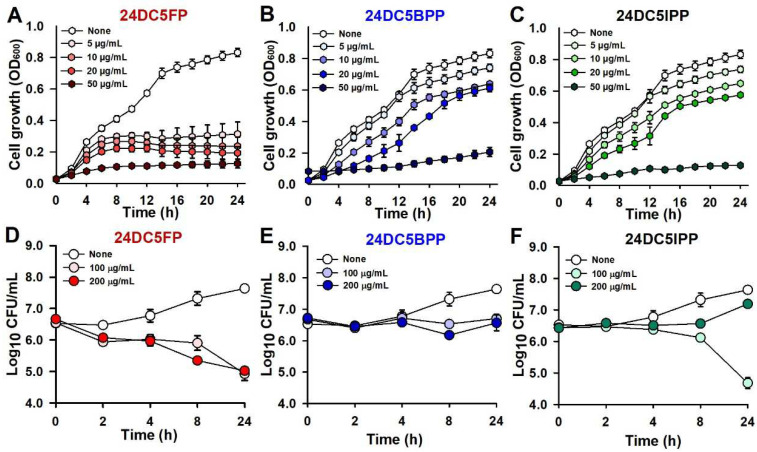
Effects of dichloro-pyrimidines on *S. aureus* planktonic cell growth. Cell growth in the presence of 24DC5FP (**A**), 24DC5BPP (**B**), and 24DC5IPP (**C**). Colony-forming unit (CFU) measurement with 24DC5FP (**D**), 24DC5BPP (**E**), and 24DC5IPP (**F**).

**Figure 3 ijms-25-12830-f003:**
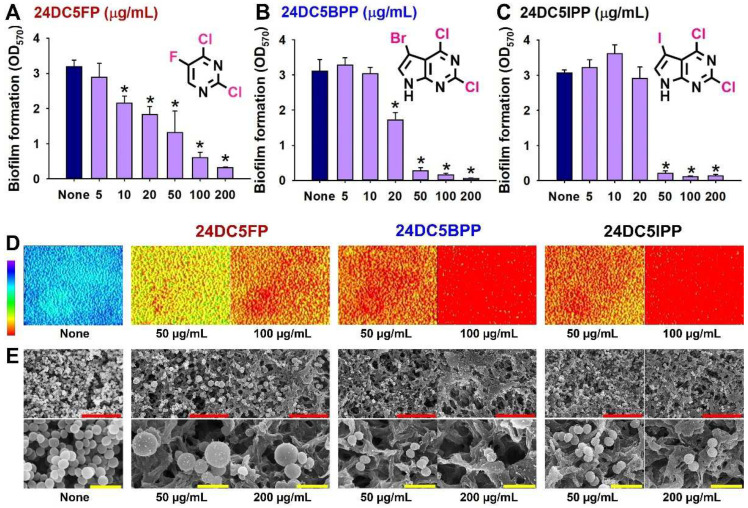
*S. aureus* biofilm inhibition by dichloro-pyrimidines. Dose-dependent inhibition by 24DC5FP (**A**), 24DC5BPP (**B**), and 24DC5IPP (**C**). 2D visualizations depict *S. aureus* biofilms treated with multi-halogenated pyrimidines (**D**). SEM analysis shows *S. aureus* biofilms exposed to multi-halogenated pyrimidines (**E**). Scale bars in red and yellow indicate measurements of 10 µm and 2 µm, respectively. * *p* < 0.05 vs. untreated controls (None).

**Figure 4 ijms-25-12830-f004:**
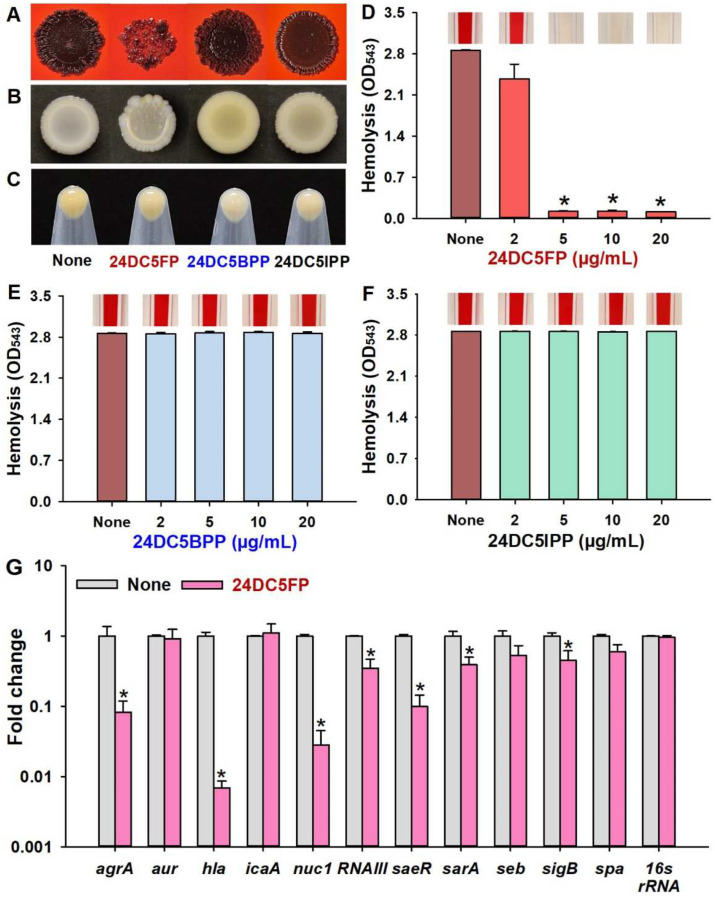
Impact of halogenated pyrimidines on *S. aureus* virulence factors and gene expression. Slime production on the Congo red agar plates (**A**), colony morphology on BHI agar plate (**B**), staphyloxanthin production (**C**), hemolytic activity (**D**–**F**), and gene expression by 24DC5FP (50 µg/mL) (**G**). * *p* < 0.05 vs. untreated controls (None).

## Data Availability

The data utilized in this study can be found within the main text of the article and the accompanying Appendix A.

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
