# Peer review of "Antibiofilm Activities of Multiple Halogenated Pyrimidines Against *Staphylococcus aureus"

_ijms, 2024, doi:10.3390/ijms252312830_

Round 1
Reviewer 1 Report
Comments and Suggestions for Authors
This study explores the antibiofilm and antimicrobial efficacy of halogenated pyrimidine derivatives against Staphylococcus aureus. Three compounds, 24DC5FP, 24DC5BPP, and 24DC5IPP, were identified as potent biofilm inhibitors. 24DC5FP reduced hemolysis by 95%, increased cell size, and suppressed gene expressions, suggesting extensive halogenation enhances the antibiofilm and antivirulence activities of these derivatives. The following comments should be address before acceptance.
1) Why the authors choose such compounds for biofilm inhibition?
2) 24DC5FP results in 95% hemolysis at 5 µg/mL concentration. High MIC value display limited activity at such lower concentration, comment this, which can probably create issue in clinical applicability.
3) What is the standard drug/antibiotic the authors use in this study? I can’t see any comparison with standard. Please clarify
4) The material methods portion should be checked for plagiarism, specially section 4.8, 4.9.
5) In minor, the presentation of the work is quite good, only few sentences should be rephrased to understand easily.
a) Aims and objective of the work line 48-53, should be rephrased with better presentation
b) Line 160-169, should be clearly presented as it’s the main issue in the manuscript, why these compounds displayed activities and why they were chosen.
c) The discussion portion should be added with the importance of antibiofilm studies, the following manuscript can be cited, if the authors found it useful
https://doi.org/10.3389/fphar.2022.890649
Comments on the Quality of English LanguageCheck for typo errors.
Author Response
Comments: This study explores the antibiofilm and antimicrobial efficacy of halogenated pyrimidine derivatives against Staphylococcus aureus. Three compounds, 24DC5FP, 24DC5BPP, and 24DC5IPP, were identified as potent biofilm inhibitors. 24DC5FP reduced hemolysis by 95%, increased cell size, and suppressed gene expressions, suggesting extensive halogenation enhances the antibiofilm and antivirulence activities of these derivatives. The following comments should be address before acceptance.
Response: We would like to thank the reviewer for thorough reading of this manuscript and insightful comments which helped us improve the manuscript's quality and scientific value. We hope our revisions have improved the manuscript to a level of the reviewer’s satisfaction.
Comments 1: Why the authors choose such compounds for biofilm inhibition?
Response 1: Thank you for your motivational question. The halogenated pyrimidine or pyrrolopyrimidine analogs were chosen based on their structural characteristics, particularly the extensive halogenation at specific positions, which has been previously reported to enhance antimicrobial and antibiofilm activities [1]. Additionally, their heterocyclic pyrimidine and pyrrolopyrimidine scaffolds are known for exhibiting diverse biological activities [2,3], making them promising candidates for biofilm inhibition. (Line 50-52) This selection aligns with our research focus on exploring the role of multi-halogenation in enhancing biofilm inhibitory properties, as demonstrated in our findings with S. aureus.
Comments 2: 24DC5FP results in 95% hemolysis at 5 µg/mL concentration. High MIC value display limited activity at such lower concentration, comment this, which can probably create issue in clinical applicability.
Response 2: Thank you for your valuable comment. The MIC value (50 µg/mL) of 24DC5FP was measured using a non-shaking method in a 96-well plate. However, for the hemolysis assay, the experiment was conducted in a shaking incubator, and in this case, even at a concentration of 5 µg/mL, cell growth was found to be low. Therefore, while the MIC value reflects the inhibitory concentration in a static environment, the hemolysis results may reflect a different impact on cell growth under shaking conditions. This result is likely due to the difference in experimental conditions (Lines 135 -137).
Comments 3: What is the standard drug/antibiotic the authors use in this study? I can’t see any comparison with standard. Please clarify
Response 3: Thank you for this important comment. As suggested, an antibiotic gentamicin has been tested for antibiofilm and anti-hemolysis activities for comparison purpose. The MIC of gentamicin against S. aureus 6538 was 20 µg/mL. As shown below, gentamicin dose-dependently inhibited S. aureus biofilm formation while gentamicin up to 100 µg/mL could not completely inhibit. Also, gentamicin dose-dependently inhibited the hemolytic activity mainly due to the growth inhibition. It appears that the antibiofilm and anti-hemolysis activities of 24DC5FP are similar to those of gentamicin. The data have been included in the Supplementary Materials.
Comments 4: The material methods portion should be checked for plagiarism, specially section 4.8, 4.9.
Response 4: Thank you for your comment. We have thoroughly reviewed the Materials and Methods section, particularly sections 4.8 and 4.9, and have revised any parts that could raise concerns regarding plagiarism. We have ensured that all references and methodologies are appropriately cited, and any similar text has been paraphrased or rewritten to meet the journal’s guidelines.
Comments 5: In minor, the presentation of the work is quite good, only few sentences should be rephrased to understand easily.
Response 5: Thank you for your valuable feedback regarding the presentation of the manuscript. As suggested, the sentences in sections a (lines 48–53), b (lines 160–169), and c (discussion portion) have been rephrased for better clarity and presentation. Additionally, an extra sentence was revised to further enhance the readability and flow of the manuscript (Lines 124-127).
Comments 5-(a): Aims and objective of the work line 48-53, should be rephrased with better presentation.
Response 5-(a): Thank you for your comments. The sentence was reorganized for better clarity in describing the methods used (Lines 52-55)
Comments 5-(b): Line 160-169, should be clearly presented as it’s the main issue in the manuscript, why these compounds displayed activities and why they were chosen
Response 5-(b): Thank you for your comment. We revised the section to clarify why these compounds exhibit specific activities and were selected. The distinction in mechanisms of action and the unique halogenation patterns (two chloride atoms at C2 and C4 and a third halogen atom) are emphasized as key factors contributing to their activity and selection.
Comments 5-(c): The discussion portion should be added with the importance of antibiofilm studies, the following manuscript can be cited, if the authors found it useful
Response 5-(c): Thank you for your valuable suggestions. As suggested, a sentence has been added to emphasize the importance of antibiofilm research. Additionally, the recommended manuscript [4] has been cited to highlight the role of enzymatic pathways in biofilm formation and inhibition. (Lines 163-164)

Reviewer 2 Report
Comments and Suggestions for Authors
The study by Shim et al. aims to evaluate various halogenated pyrimidines as potential antibiofilm agents against Staphylococcus aureus. While the study presents interesting findings, a few points could be addressed to strengthen the manuscript:
1. Line 26-27: The authors should provide appropriate references to support the statements made here.
2. Section 2.4 (Impact of 24DC5FP on Gene Expression in S. aureus): The authors should include references for the primer sequences used in the study.
3. Line 259: A reference should be provided to support the description of the Scanning Electron Microscopy sample preparation method.
4. Lines 328-329: The authors should consider expanding this section to describe potential future approaches or methods to elucidate the mode of action of the halogenated pyrimidines.
5. Line 168-169: The statement "While speculative, increased halogenation in their structures could further enhance their activity" should be supported by a reference, if available.
Author Response
Comments: The study by Shim et al. aims to evaluate various halogenated pyrimidines as potential antibiofilm agents against Staphylococcus aureus. While the study presents interesting findings, a few points could be addressed to strengthen the manuscript:
Response: We would like to thank the reviewer for thorough reading of this manuscript and insightful comments which helped us improve the manuscript's quality and scientific value. We hope our revisions have improved the manuscript to a level of the reviewer’s satisfaction.
Comments 1: Line 26-27: The authors should provide appropriate references to support the statements made here.
Response 1: Thank you for your comment. As suggested, two more related reference [5,6] have been added while the statements made in lines 27-28 regarding Staphylococcus aureus and its biofilm formation are supported by the reference [7].
Comments 2: Section 2.4 (Impact of 24DC5FP on Gene Expression in S. aureus): The authors should include references for the primer sequences used in the study.
Response 2: Thank you for your comment. The primer sequences used in this study were obtained from previously published literature, as mentioned in the Methods section. To ensure clarity, we have re-emphasized this information in Section 2.4 by explicitly stating that "The primer sequences used in this study were adopted from a previously published study [22] " We hope this addition addresses the reviewer’s concern.
Comments 3: Line 259: A reference should be provided to support the description of the Scanning Electron Microscopy sample preparation method.
Response 3: Thank you for your comment. A reference has been added to support the detailed description of the Scanning Electron Microscopy (SEM) sample preparation method. The reference [8] is now included in the relevant section of the manuscript, and we hope this addition addresses the reviewer’s concern.
Comments 4: Lines 328-329: The authors should consider expanding this section to describe potential future approaches or methods to elucidate the mode of action of the halogenated pyrimidines.
Response 4: Thank you for your valuable comment. To address your suggestion, we have expanded the section to include potential future approaches for elucidating the mode of action of halogenated pyrimidines (Lines 351 -355). We hope this addition satisfies the reviewer’s suggestion.
Comments 5: Line 168-169: The statement "While speculative, increased halogenation in their structures could further enhance their activity" should be supported by a reference, if available.
Response 5: Thank you for your comment. We have revised the sentence to clarify that the statement "increased halogenation in their structures could further enhance their activity" is based on the patterns observed in our study and is a speculative suggestion. While we were unable to find a direct reference to fully support this hypothesis, we believe that the structure-activity relationship demonstrated in our findings supports this reasoning (Lines 185-187). We hope this clarification addresses the reviewer’s concern.

Reviewer 3 Report
Comments and Suggestions for Authors
The manuscript “Antibiofilm activities of multiple halogenated pyrimidines against Staphylococcus aureus” describes the selection and assays performed with some halogenated pyrimidine derivatives in order to prove their antibiofilm abilities against one of the most critical pathogens that have propensity to develop such complex communities, that are highly resistant to both antibiotics and host immune responses. The paper is well organised, all data were proper collected and clear and detailed presented. Moreover, the antibiofilm activity was related to haemolytic activity and some suppression of the gene expressions. The literature is comprehensive and updated. My overall comment is that these data present interest concerning the good activity against antibiofilm developed by S. aureus, demonstrated for three of selected derivatives. As result, in my opinion paper can be accepted in current form.
Author Response
Comments: The manuscript “Antibiofilm activities of multiple halogenated pyrimidines against Staphylococcus aureus” describes the selection and assays performed with some halogenated pyrimidine derivatives in order to prove their antibiofilm abilities against one of the most critical pathogens that have propensity to develop such complex communities, that are highly resistant to both antibiotics and host immune responses. The paper is well organised, all data were proper collected and clear and detailed presented. Moreover, the antibiofilm activity was related to haemolytic activity and some suppression of the gene expressions. The literature is comprehensive and updated. My overall comment is that these data present interest concerning the good activity against antibiofilm developed by S. aureus, demonstrated for three of selected derivatives. As result, in my opinion paper can be accepted in current form.
Response: Thank you for taking the time to review our manuscript and for your positive evaluation and valuable feedback. We are delighted that you found the structure of the manuscript and the clarity and detail of the data presentation to be commendable. Additionally, we greatly appreciate your interest in the significant potential demonstrated by the three selected derivatives in combating Staphylococcus aureus biofilm formation. We are proud that this manuscript reflects your insightful comments and suggestions, and we sincerely thank you again for your favorable assessment and recommendation for acceptance in its current form.

Round 2
Reviewer 1 Report
Comments and Suggestions for Authors
The authors have revised the manuscript well.